# Methylation profiling identifies two subclasses of squamous cell carcinoma related to distinct cells of origin

Manuel Rodríguez-Paredes[1], Felix Bormann[1], Günter Raddatz[1], Julian Gutekunst[1], Carlota Lucena-Porcel[2,3], Florian Köhler[1], Elisabeth Wurzer[4], Katrin Schmidt[4], Stefan Gallinat[4], Horst Wenck[4], Joachim Röwert-Huber[5], Evgeniya Denisova[6], Lars Feuerbach[6], Jeongbin Park[7], Benedikt Brors [6,8,9], Esther Herpel[2,3], Ingo Nindl[5], Thomas G. Hofmann[1], Marc Winnefeld[4] & Frank Lyko [1]

Cutaneous squamous cell carcinoma (cSCC) is the second most common skin cancer and usually progresses from a UV-induced precancerous lesion termed actinic keratosis (AK). Despite various efforts to characterize these lesions molecularly, the etiology of AK and its progression to cSCC remain partially understood. Here, we use Infinium MethylationEPIC BeadChips to interrogate the DNA methylation status in healthy, AK and cSCC epidermis samples. Importantly, we show that AK methylation patterns already display classical features of cancer methylomes and are highly similar to cSCC profiles. Further analysis identifies typical features of stem cell methylomes, such as reduced DNA methylation age, non-CpG methylation, and stem cell-related keratin and enhancer methylation patterns. Interestingly, this signature is detected only in half of the samples, while the other half shows patterns more closely related to healthy epidermis. These findings suggest the existence of two subclasses of AK and cSCC emerging from distinct keratinocyte differentiation stages.

[1] Division of Epigenetics, DKFZ-ZMBH Alliance, German Cancer Research Center, 69120 Heidelberg, Germany. [2] Institute of Pathology, Heidelberg University, 69120 Heidelberg, Germany. [3] Tissue Bank of the National Center for Tumor Diseases (NCT), 69120 Heidelberg, Germany. [4] Research & Development, Beiersdorf AG, 20253 Hamburg, Germany. [5] Department of Dermatology, Venereology and Allergy, Charité, University Hospital, 10117 Berlin, Germany. [6] Division of Applied Bioinformatics, German Cancer Research Center (DKFZ), 69120 Heidelberg, Germany. [7] Division of Theoretical Bioinformatics, German Cancer Research Center (DKFZ), 69120 Heidelberg, Germany. [8] National Center for Tumor Diseases (NCT), 69120 Heidelberg, Germany. [9] German Cancer Consortium (DKTK), German Cancer Research Center (DKFZ), 69120 Heidelberg, Germany. Correspondence and requests for materials should be addressed to F.L. (email: f.lyko@dkfz.de)

Cutaneous squamous cell carcinoma (cSCC) is a non-melanoma skin cancer type that originates from epidermal keratinocytes and represents about 20% of all skin cancers, with up to 700 000 new cases annually diagnosed in the USA[1]. Although usually treated effectively by surgery and/or radiotherapy, the percentage of cSCC tumors that recur or metastasize within 5 years is 8% and 5%, respectively[1]. Chronic sun damage is known to be the main cause of cSCC, which preferably arises on highly exposed areas such as face and neck, from a precancerous lesion termed actinic keratosis (AK)[2]. AK is defined as an ultraviolet light (UV)-induced keratinocytic dysplasia with abnormal cells[3]. Although most AKs do not progress to cSCC and many regress spontaneously, the annual risk of progression is estimated at 0.025–20% per individual lesion[4]. The precise causes of the AK to cSCC transition still remain obscure. It has been shown that cSCC displays a higher mutational load than AK[5]. However, AK samples already harbor mutations in the same genes that are found mutated in metastatic and aggressive cSCC, such as *TP53*, *NOTCH1-2*, *FAT1*, or *MLL2*[5]. Other typical cancer-associated genes that are known to be likewise mutated in both entities are *CDKN2A*, *RAS*, *EGFR*, or *MYC*[6–10]. Molecular studies using array-based expression profiling have been largely inconclusive[11–15]. However, a recent study that used RNA-seq to analyze and compare human samples with an UV-irradiated hairless mouse model, identified distinct transcriptional networks driving the initial emergence of AK from healthy epidermis and the transition from AK to cSCC[5].

DNA methylation is a covalent epigenetic modification of cytosines within CpG dinucleotides[16,17]. The modification is established and maintained by a set of specific enzymes called DNA methyltransferases and regulates cellular identity through the modulation of gene expression[16,17]. Healthy cells are characterized by the absence of DNA methylation at CpG islands, short CpG-rich regions present in ~60% of human promoters, and the extensive methylation of gene bodies and repetitive regions[18]. The deregulation of the normal methylome is a major hallmark of human cancers[19] and frequently characterized by global hypomethylation of lamina-associated domains (LAD) as well as widespread CpG island promoter hypermethylation[19,20]. Importantly, altered DNA methylation patterns emerge early in tumorigenesis[21], and can be used as biomarkers for tumor detection, diagnosis and prognosis[22,23].

Our current knowledge about epigenetic changes associated with cSCC is very limited and mostly comprises a moderate number of cancer-associated genes that become silenced by CpG island promoter hypermethylation. Examples include *CDKN2A*, *CDH1*, *DAPK1*, or *MGMT*[24,25]. In addition, a recent study suggested that UV-irradiation through chronic sun exposure gives rise to large hypomethylated blocks in healthy epidermis, and that

these blocks are conserved in cSCC[26]. Finally, with respect to the transition between AK and cSCC, it has been suggested that *CDH1* promoter hypermethylation might increase from normal skin to AK and cSCC[27], and that the silencing of miR-204 might play a role in the progression from AK to cSCC[28].

Here, we investigate DNA methylation changes in the progression from healthy epidermis to AK and cSCC using Infinium MethylationEPIC BeadChips, which contain about 850,000 CpG probes. Our results provide the most comprehensive epigenomic analysis of cSCC development to date and suggest the existence of two distinct subclasses that reflect different cell-of-origin differentiation stages.

## Results

**AK and cSCC show similar aberrant methylation patterns.** Epigenetic modification patterns show a considerable degree of cell-type specificity, which represents a major confounding factor for the interpretation of epigenetic data. For the analysis of human AK and cSCC methylomes, we therefore placed a particular emphasis on the cell-type purity of our samples. Consequently, we separated the epidermal from the dermal parts of punch biopsies and exclusively included samples where a sufficiently large area of the epidermal layer could be dissected in order to isolate at least 500 ng of genomic DNA. In addition, all biopsies were taken from the center of the lesions to ensure the histopathological distinction of AK and squamous cell carcinoma in situ (see Methods for details). Altogether, we analyzed 12 normal epidermis samples, 16 AK epidermis samples, and 18 cSCC epidermis samples using Infinium 850k methylation arrays (Supplementary Table 1). The resulting data sets were quality-assessed, functionally normalized, filtered for potentially confounding probes (those located in sex chromosomes, SNP-containing, and self-hybridizing), and subsequently analyzed using Minfi[29].

Principal component analysis (PCA) performed on all 850k probes clearly separated the normal epidermis from AK and cSCC samples, but also indicated highly overlapping patterns between AK and cSCC (Fig. 1a). Moreover, while normal epidermis samples grouped very homogeneously, AK and cSCC distributed more heterogeneously. Pairwise comparisons of the methylation patterns from the three sample groups using Minfi revealed a high number (372,213) of significantly (adjusted $P < 0.05$, $F$-test) differentially methylated probes between AK and normal epidermis (Fig. 1b). A similarly high number (310,102) of differentially methylated probes was detected when cSCC and normal samples were compared (Fig. 1b). However, in agreement with our PCA, no significantly differentially methylated probes were detected between AK and cSCC (Fig. 1c). These findings establish pronounced methylation differences between healthy

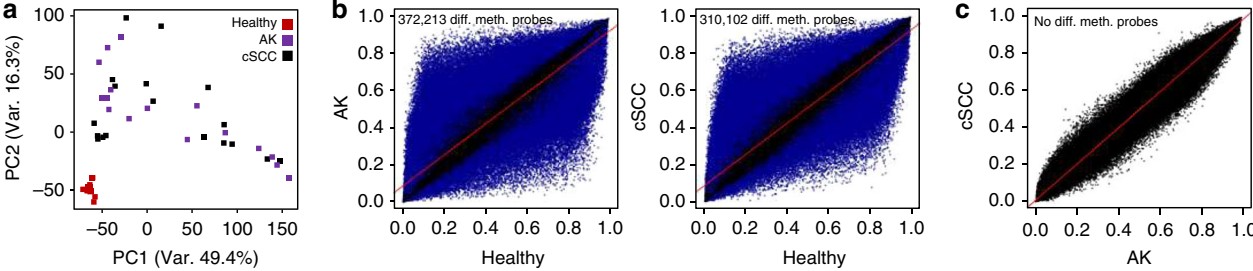

**Fig. 1** Actinic keratosis (AK) and cutaneous squamous cell carcinoma (cSCC) show similar aberrant methylation patterns. **a** Principal component analysis of 12 healthy, 16 AK, and 18 cSCC epidermis samples using all 850k CpG probes. **b** Scatter plots comparing the epidermis methylomes of healthy and AK samples (left panel), and healthy and cSCC samples (right panel). Differentially ($P < 0.05$, $F$-test) methylated probes are shown in blue. **c** Scatter plot comparing the epidermis methylomes of AK and cSCC samples. No significantly ($P < 0.05$, $F$-test) differentially methylated probes were detected

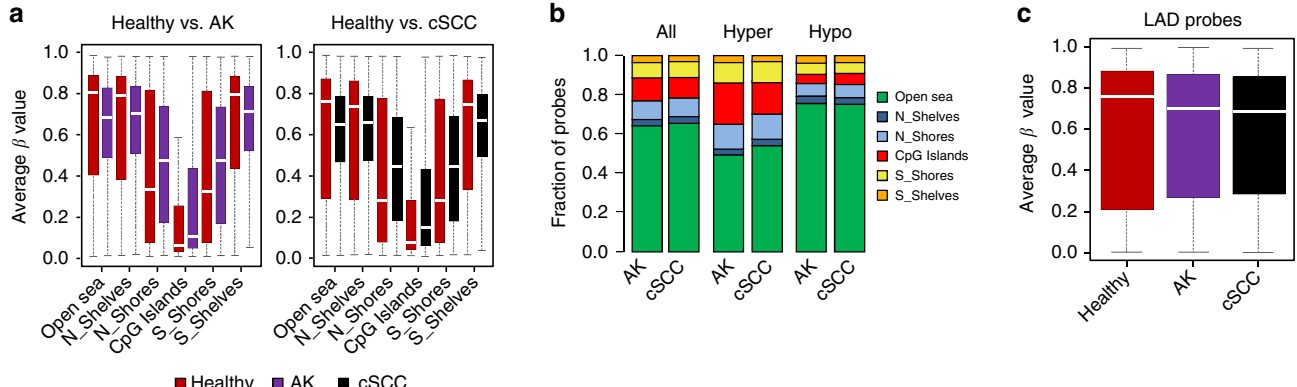

**Fig. 2** AK and cSCC methylomes show cancer-specific features. **a** Methylation status of the different epigenomic substructures in the epidermis of AK (left) and cSCC (right) patients compared to healthy controls. The box plots indicate highly significant ($P \leq 9.1E-77$, two-sided $t$-test) hypermethylation of the CpG islands and hypomethylation of the Open Sea probes in both AK and cSCC samples. **b** Fractions of hyper and hypomethylated CpGs in AK and cSCC epidermis, within different epigenomic substructures and in comparison to healthy skin. **c** Probes within lamina-associated domains (LADs) are significantly ($P \leq 3.4E-288$, two-sided $t$-test) hypomethylated in AK and cSCC when compared to healthy epidermis

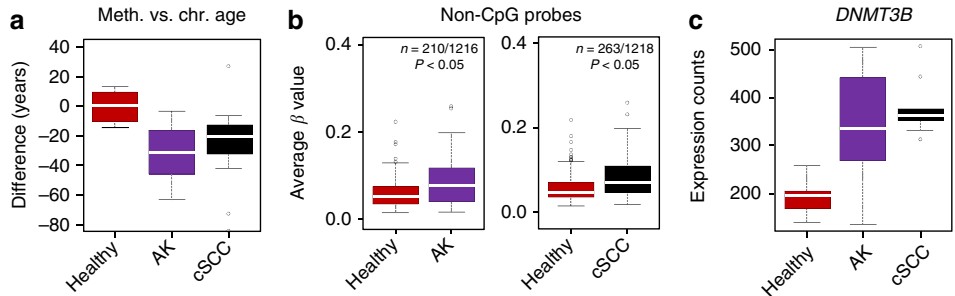

**Fig. 3** AK and cSCC display typical features of stem cell methylomes. **a** Mean difference between chronological and methylation-based biological age for healthy, AK, and cSCC samples. **b** Box plots indicating the average methylation levels of the 210 (left panel) and 263 (right panel) differentially ($P < 0.05$, $F$-test) methylated non-CpG probes in AK and cSCC. **c** The DNA methyltransferase *DNMT3B* gene is significantly upregulated in AK ($P < 0.002$, two-sided $t$-test) and cSCC ($P < 3.43E-06$, two-sided $t$-test). RNA-seq gene expression data from 7 healthy, 10 AK, and 9 cSCC epidermis samples was obtained from Chitsazzadeh et al.[5]

and diseased (AK, cSCC) epidermis samples and also suggest substantial epigenetic similarities between the precancerous lesions and the tumor samples.

**AK and cSCC methylomes show cancer-specific features.** Cancer methylomes are often characterized by specific features, such as hypomethylated lamina-associated domains (LADs) and hypermethylated CpG islands[19,20]. Furthermore, hypomethylated LADs have also been described as a prominent feature of cSCC[26]. We therefore compared the methylation status of different epigenomic substructures such as CpG islands, shelves, shores, and open sea regions in our data sets. The results showed robust hypomethylation of open sea probes as well as hypermethylation of CpG islands and their shores in both AK and cSCC samples. (Fig. 2a). Similarly, the fraction of probes changing their methylation in the different epigenomic substructures was also very similar between normal epidermis and AK or cSCC (Fig. 2b). Finally, we used published data sets[30] to define the association of the 850k probeset with LADs. This revealed that LAD probes were significantly hypomethylated in AK and in cSCC (Fig. 2c). Our results thus suggest that the premalignant AK samples already display key features of cancer methylomes and that these features are conserved in cSCC.

**AK and cSCC display typical stem cell methylation features.** DNA methylation can be used to predict the chronological age of a given human tissue with high accuracy[31]. When we applied the age predictor to our samples, we observed a methylation age for the AK and cSCC samples that was clearly below the chronological age of the respective patients (Fig. 3a). This effect is reminiscent of the reduced methylation age observed for stem cells[31]. Another key feature of stem cells is the presence of DNA methylation marks outside of the canonical CpG context (non-CpG methylation)[32,33]. Indeed, our analysis revealed significantly ($P < 0.05$, $F$-test) increased non-CpG methylation levels, both in AK and in SCC samples (Fig. 3b). Non-CpG methylation has been closely associated with the DNMT3B DNA methyltransferase, which also represents an important epidermal stem cell gene[34]. Consistent with our methylation data, we observed a significant increase of *DNMT3B* expression in AK and cSCC samples when compared to healthy controls (Fig. 3c). Together, these results suggest that DNA methylation patterns of AK and cSCC show key features of stem cell methylomes.

Stem cell features of AK and cSCC were also confirmed by immunohistochemical analysis of an independent sample set (11 healthy skin, 11 AK, and 11 cSCC). Specific markers included p63, an epidermal stem cell transcription factor[35,36], as well as keratins K5 and K14, that are predominantly expressed in basal keratinocytes[37,38]. Our results confirmed expression of these three markers in the epidermal basal layers of healthy skin samples (Fig. 4), while atypical keratinocytes from AK and cSCC samples showed broad p63, keratin K5, and keratin K14 expression

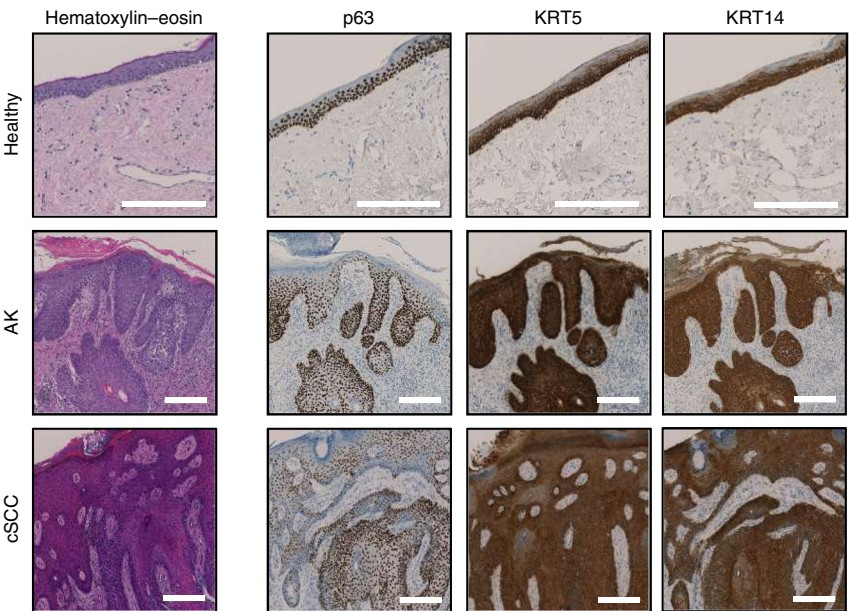

**Fig. 4** AK and cSCC express epidermal stem cell markers. The figure shows representative hematoxylin and eosin stainings, as well as p63, keratin K5, and keratin K14 immunostainings in epidermis from healthy donors, AK and cSCC patients, respectively. Ectopic expression of the three epidermal stem cell markers was observed in AK and cSCC samples. Scale bars, 200 µm

(Fig. 4). These findings are consistent with previous reports that had shown elevated expression levels of epidermal stem cell markers in AK or cSCC[39–42], suggesting that AK and cSCC can arise from mutated epidermal stem cells of the epidermal basal layer[43].

**Keratin gene methylation defines two subclasses of AK/cSCC.** Keratin expression patterns correspond to the epithelial cell type and its degree of differentiation[44]. We therefore analyzed the methylation patterns of the keratin gene clusters on chromosomes 12 and 17 and observed major DNA methylation differences between healthy donors and AK and cSCC patients (Fig. 5a, Supplementary Figure 1). A closer inspection of individual keratin genes confirmed this notion and also indicated the presence of two distinct keratin methylation patterns within the AK and cSCC sample groups (Fig. 5b). Intriguingly, the analysis of *TP63* gene methylation provided a similar result (Supplementary Figure 2), suggesting that methylation differences within sample groups may be differentiation-related.

For a more comprehensive and systematic keratin gene methylation pattern analysis, we extracted the methylation data for all 1364 probes that are associated with the 55 keratin genes. This information was also extracted from 7824 cancer data sets from The Cancer Genome Atlas (TCGA), representing 26 different tumor entities. In agreement with our initial findings, principal component analysis again distributed the AK and cSCC samples into two groups (Fig. 5c). Interestingly, one group clustered together with the cancer samples, while the other group was closely related to the healthy epidermis samples (Fig. 5c). Similarly, hierarchical clustering of the same samples and tumor entities also separated the AK/cSCC samples into own branches, one closely related to healthy epidermis and the other one related to other epithelial tumor entities such as kidney, thyroid, or breast cancer (Fig. 5d). In conclusion, keratin gene methylation patterns clearly define two distinct subtypes of AK and cSCC that potentially originate from different keratinocyte differentiation stages.

**Epidermal differentiation stages define AK/cSCC subclasses.** The Infinium MethylationEPIC array provides unique opportunities for the analysis of enhancer methylation states and enhancers have been shown to be a particularly relevant target of DNA methylation[32,45]. Furthermore, enhancer methylation patterns can be used to define cellular identity[46]. To determine whether the observed AK/cSCC subtypes could arise from distinct keratinocyte differentiation stages, we classified our AK and cSCC epidermal samples according to the DNA methylation status of 104,477 probes located in the 77,154 enhancers of H1 human embryonic stem cell (ESC) line, as well as 123,257 probes located in the 79,155 enhancers of normal human keratinocytes[47,48] (Fig. 6a). Subsequent hierarchical clustering identified the same subclasses of AK and cSCC: one closely related to keratinocytes and the other one related to ESCs (Fig. 6b). These results strongly suggest that the two observed AK/cSCC subclasses develop from two distinct cell types of origin that are related to undifferentiated epidermal stem cells and to more differentiated keratinocytes, respectively.

We further refined our analysis by the integration of reference data sets, using published methylation profiles of epidermal stem cells (EpSC) and differentiated keratinocytes[34]. Unsupervised clustering of the corresponding enhancer methylation profiles again separated the AK and SCC samples into the two previously described subclasses (Fig. 6c). In addition, our results also identify the EpSC-specific enhancers as a key feature of the EpSC-like subgroup (Fig. 6c). Pathway Analysis of the genes ($n = 1188$) associated with the hypermethylated EpSC-specific enhancer regions highlighted their role in developmental functions (Supplementary Figure 3). Finally, we also performed whole-exome sequencing of 20 samples (10 AK and 10 cSCC; Supplementary Table 1) to investigate a potential relationship between DNA methylation-based subtypes and genetic mutation patterns. The results showed a higher mutational heterogeneity in the AK samples (Supplementary Figure 4), as described previously[5]. However, no correlation between genetic mutations and the DNA methylation patterns defining the two subgroups could be found (Supplementary Figure 4). Collectively, these findings confirm the existence of two distinct subclasses of AK and cSCC, which originate from different keratinocyte differentiation stages and are independent of the mutational profile.

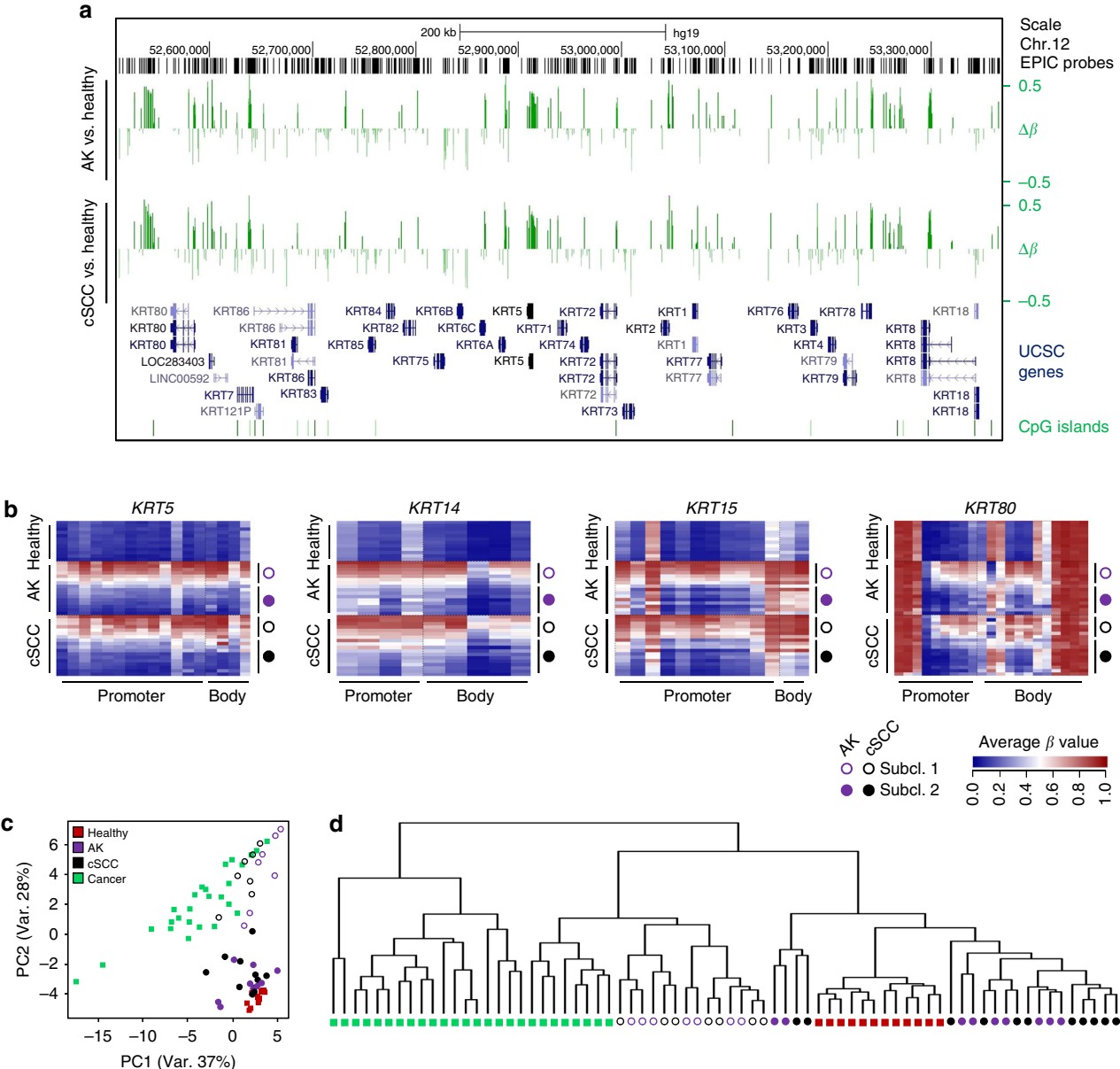

**Fig. 5** A specific methylation signature at keratin gene clusters identifies two distinct subclasses of AK/cSCC. **a** UCSC genome browser tracks showing significant DNA methylation differences ($\Delta\beta$, $P < 0.05$, $F$-test, vertical green lines) at the keratin gene cluster on chromosome 12 for AK (upper part) and cSCC (lower part) in comparison to healthy epidermis. **b** Specific DNA methylation patterns of the genes encoding keratins K5, K14, K15, and K80 (*KRT5*, *KRT14*, *KRT15*, and *KRT80*, respectively). Heatmaps show DNA methylation levels (in $\beta$ values, from blue (= 0) to red (= 1)) of probes (columns) located in the promoter and gene body, for individual healthy, AK, and cSCC samples (rows). Filled and empty circles denote the two distinct subclasses of AK (*purple*) and cSCC (*black*), respectively. **c** Principal component analysis of healthy, AK, and cSCC epidermis samples based on 1364 keratin-associated methylation probes. For comparisons, keratin methylation patterns from 26 additional tumor entities were also included. **d** Hierarchical clustering of the dataset shown in **c**. The 26 tumor entities depicted in this graph, and also used in **c**, are from left to right: testicular germ cell tumors, liver hepatocellular carcinoma, esophageal carcinoma, head and neck squamous cell carcinoma, lung squamous carcinoma, cervical squamous cell carcinoma and endocervical adenocarcinoma, uterine corpus endometrial carcinoma, uterine carcinosarcoma, lung adenocarcinoma, stomach adenocarcinoma, ovarian serous cystadenocarcinoma, bladder urothelial carcinoma, colon adenocarcinoma, rectum adenocarcinoma, skin cutaneous melanoma, adrenocortical carcinoma, glioblastoma multiforme, sarcoma, thyroid carcinoma, kidney renal papillary cell carcinoma, cholangiocarcinoma, prostate adenocarcinoma, breast invasive carcinoma, pancreatic adenocarcinoma, kidney renal clear cell carcinoma, and mesothelioma

## Discussion

Altered DNA methylation patterns are considered a classical hallmark of cancer[19,49], but their precise significance and functional relevance for tumorigenesis are still not completely understood. We have now generated high-resolution methylation profiles to analyze for the first time the methylomes of cSCC and its precursor lesion, AK, in epidermis samples that were carefully prepared to avoid contamination from surrounding tissue. This focused our analysis on keratinocyte-related methylation patterns and minimized the impact of confounding patterns from other cell types. Our results classify cSCC into two distinct subgroups that are defined by stem cell-like and keratinocyte-like methylation patterns, respectively. Interestingly, the same subclassification could be applied to the AK samples. These findings are

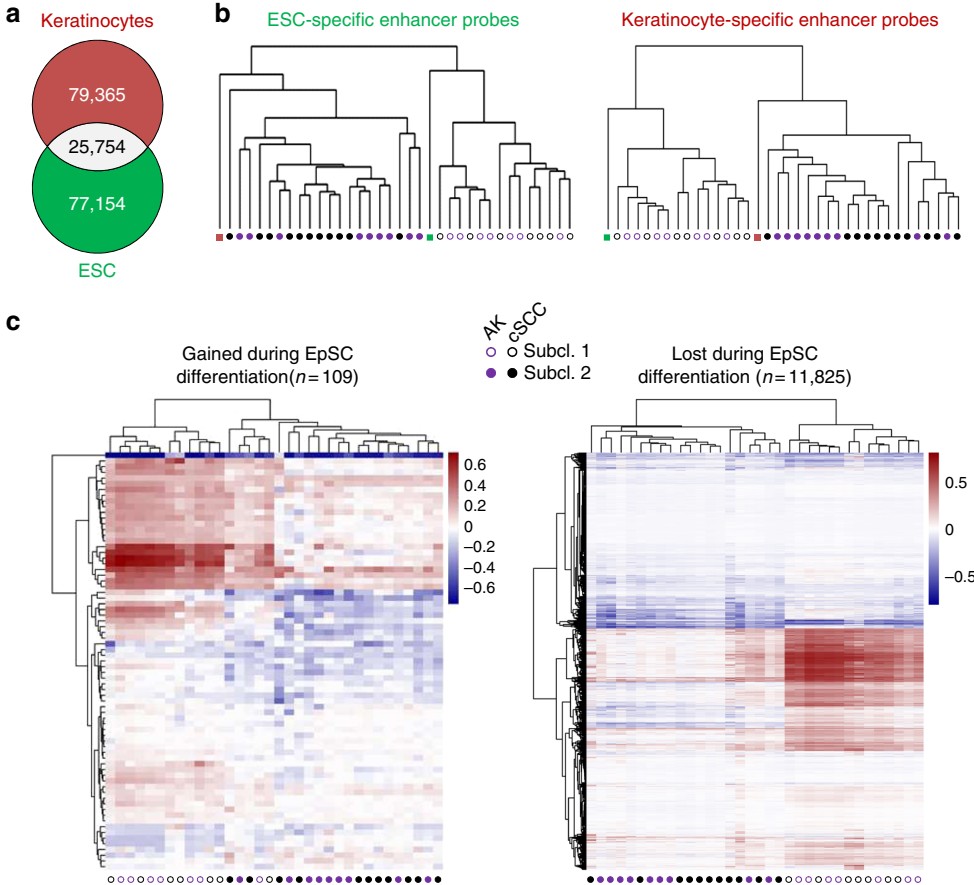

**Fig. 6** Enhancer methylation patterns separate AK and cSCC into keratinocyte-like and epidermal stem cell-like subtypes. **a** Venn diagram displaying the total number of enhancer regions defined for human embryonic stem cells and keratinocytes, respectively[47, 48]. As expected, both cell types only share about a quarter of their enhancers (**b**) Hierarchical clustering of AK and cSCC epidermis samples based on the methylation status of the 850k probes contained in ESC (left) and keratinocyte (right) enhancers. Methylation profiles of ESCs epidermis samples were used as reference. Filled and empty circles denote the two distinct subclasses of AK (purple) and cSCC (black), respectively. **c** Heatmaps showing the methylation profiles of enhancers gained (n = 109, left) and lost (n = 11,825, right) during EpSC differentiation, in AK and cSCC epidermis samples, respectively. Colors represent the normalized average methylation levels of each enhancer region

consistent with the notion that cSCC develops from AK and that both lesions share common cell types of origin.

Whether AK represents a benign precancerous lesion or malignant tissue is a matter of ongoing debate[4]. Our observation that AK methylomes display typical cancer-related features, such as CpG island hypermethylation and LAD hypomethylation may indicate a significant malignant potential of AK. This is consistent with previous reports showing that both AK and cSCC share mutations in key genes associated with cancer development and progression[5–10]. In addition, several published studies failed to identify major expression differences between AK and cSCC[5,11–15]. This is in agreement with our finding that epigenetic differences in comparison to normal skin are shared between unstratified groups of AK and cSCC samples.

Our detailed analysis of the AK and cSCC methylomes also revealed key features of stem cell methylomes. In particular we observed a lower methylation age and a higher level of non-CpG methylation, both typical characteristics of stem cells[31–33]. These results, together with the ectopic expression of the epidermal stem cell markers[50] p63, keratin K5 and keratin K14, can be interpreted to reflect the development of cSCC from progenitor cells in the epidermal basal layer, as previously proposed[43].

Cancer-associated DNA methylation profiles are increasingly interpreted to reflect the epigenetic program of the cancer cell-of-origin[51–53]. This concept is similar to the use of keratin expression patterns for tumor classification, which is based on the finding that epithelial tumors largely maintain the keratin expression pattern of their cell-of-origin[54]. Interestingly, our analysis revealed two distinct keratin methylation profiles within our AK/cSCC sample set, one resembling that of the healthy epidermis and the other one resembling that of other tumor entities, which often present a certain degree of dedifferentiation[55,56]. These results strongly suggested two different cell types of origin for the two observed AK/cSCC subgroups. Our analysis of enhancer methylation patterns further associated these two cell types with epidermal stem cells and more differentiated keratinocytes, respectively. Altogether, our findings support a model (Fig. 7) where AK and cSCC originate from two (or more) differentiation stages of epidermal stem cells. The detailed characterization of these cell types and the analysis of their clinical significance will be important aspects for future studies.

## Methods
**Samples.** To obtain healthy epidermis samples, suction blisters[57] were induced on the forearms of healthy male volunteers. After their removal, suction blister roofs were immediately stored at −80 °C. Suction blistering was approved by the Freiburger Independent Ethics Committee (011/1973) and written, informed patient consents were given from all volunteers. AK and cSCC samples were obtained as punch biopsies (diameter 4 mm) at the Charité University Hospital (Berlin, Germany) from three diagnostic stages of AK and cSCC. Half of the tissue was immersed in liquid nitrogen within 2 min of resection and stored at −70 °C. The

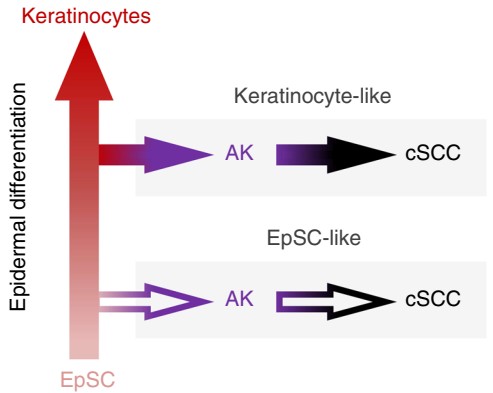

**Fig. 7** Model describing the emergence of AK/cSCC subtypes from different stages of epidermal differentiation. Mutations giving rise to AK can result in the transformation of distinct epidermal differentiation stages, resulting in two distinct subclasses of AK/cSCC

other half of each biopsy (excluding normal skin specimens) was fixed in formalin, embedded in paraffin, and sections were stained with hematoxylin and eosin for histology. To exclude a misdiagnosis of squamous cell carcinoma in situ (SCCIS) as AK, punch biopsies were taken from the center of the lesions. Biopsied tissues were further dissected to allow histopathological analyses of the core of the lesions. AK samples with histopathological features of SCCIS were excluded from further analysis. The study was approved by the local ethics committee at the Charité, University Hospital, Berlin, Germany (number Si. 248). An overview of all samples is provided in Supplementary Table 1. The epidermal parts were separated from the dermal parts of the punch biopsies by heat split (56 °C for 2 min) and careful manual dissection. DNA was isolated using the QIAamp DNA Investigator Kit (Qiagen).

**Analysis of Infinium 850k data**. DNA methylation profiles were obtained using Infinium MethylationEPIC BeadChips (Illumina) according to the manufacturer's protocols. Methylation analysis was performed using the R Bioconductor package Minfi (v1.20.2)[29]. In short, raw IDAT files were read and preprocessed. Methylation loci (probes) were filtered for high detection $p$-value ($P > 0.01$, as provided by Minfi[29]), location on sex chromosomes, ability to self-hybridize, and potential SNP contamination. Array normalization was carried out using the *preprocessFunnorm* function, available in Minfi[58]. Quality control was performed after every pre-processing step. Differentially methylated probes were identified by fitting a linear model followed by statistical analysis using an empirical Bayes method to moderate standard errors. Eventually, differentially methylated probes were filtered by significance threshold ($P < 0.05$, $F$-test, after correction for multiple testing using the Benjamini–Hochberg method). Methylation age was determined by using the DNA methylation calculator (https://dnamage.genetics.ucla.edu/). LAD association of Infinium 850k probes was determined using previously published data sets[30].

**Gene expression analysis**. RNA-seq data were extracted from a published data set[5]. We plotted the normalized expression counts for healthy, AK, and cSCC samples using R. $P$-values were calculated by a $t$-test and corrected for multiple testing (Benjamini–Hochberg method).

**Immunohistochemistry (IHC) analysis**. IHC analysis was performed on 1 μm-thick sections of formalin-fixed, paraffin-embedded (FFPE) epidermis from 11 healthy donors, 11 AK, and 11 cSCC patients, respectively. All samples shared the same gender (male) and ethnicity (Caucasian), and were provided and processed by the tissue bank of the National Center for Tumor Diseases (NCT, Heidelberg, Germany) in accordance with its own regulations and with the approval of the ethics committee of the University of Heidelberg. Sections were cut using a HMA 340E Electronic Rotary Microtome (Thermo Scientific), dried and stored at room temperature until their analysis with a BenchMark ULTRA instrument (Ventana Medical Systems). Antigens were retrieved with Protease I (Ventana Medical Systems) for 12 min at room temperature. Primary antibodies and dilutions used to detect the different markers were as follows: p63 (790–4509, Ventana Medical Systems; 0.140 μg mL$^{-1}$), keratin K5 (790–4554, Ventana Medical Systems; 10.4 μg mL$^{-1}$) and keratin K14 (760–4805, Ventana Medical Systems; 1–5 μg mL$^{-1}$). Sections were incubated with the corresponding antibodies at 36 °C for 24 min and detected using the OptiView DAB IHC Detection Kit (Ventana Medical Systems). A NanoZoomer Slide Scanner (Hamamatsu) was used to generate the final images and their analysis was performed with the Aperio ImageScope viewing software (Leica Biosystems, version 12.3.2.8013).

**Keratin gene methylation analysis**. Keratin gene methylation analyses were based on the 1364 850k probes contained in 55 keratin genes. For comparisons, we also obtained Infinium HumanMethylation450 BeadChip level 3 methylation data from 7824 samples of TCGA, and averaged them for each of the 26 considered cancer types. After extracting the 522 450k CpG probes contained in all keratin genes, the $\beta$ values corresponding to each gene were also averaged. Finally, PCA and unrooted cluster dendrograms were performed using the R packages FactoMineR and APE, respectively.

**Enhancer methylation analysis**. ChromHMM segmentations for H1 ESCs and normal human epithelial keratinocytes were downloaded from the University of California Santa Cruz (UCSC) web server (https://genome.ucsc.edu) and enhancer regions were extracted (77,154 and 79,365, respectively). After identifying the 850k CpG probes contained in both enhancer sets (104,477 and 123,257, respectively), the methylation values of those probes contained in each enhancers were averaged for all individual AK and cSCC samples. Methylation data for H1 cells were obtained from the Roadmap Epigenomics Project[59], methylation data for keratinocytes was obtained from the healthy epidermis samples. Hierarchical clustering was performed using APE. Locations of enhancers gained and lost during EpSc differentiation were obtained from published data[34]. We identified the 850k probes contained in these regions (387 and 85,381, respectively) and averaged the methylation values of those contained in each of the enhancers for all our samples. After subtracting the $\beta$ values of healthy epidermis, the final numbers were used for sample clustering and visualized as heatmaps.

**Pathway analysis**. Ingenuity Pathway Analysis software (Qiagen) was used to assess the developmental role of the genes ($n = 1188$) contained in the EpSC-specific enhancer regions found hypermethylated in the EpSC-related AK/cSCC subclass (normalized $\Delta\beta$ value > 0.2).

**Whole-exome sequencing**. Exome capture was performed using the Agilent SureSelect Human All Exon v5 kit. The capture area comprised 357,999 exons from 21,522 genes (~50 Mb in total). Paired-end 100 bp DNA sequencing reads were subsequently generated on a HiSeq 4000 system (Illumina), achieving an average coverage of 180 × (Supplementary Table 2). Data preprocessing was performed using the One Touch Pipeline (OTP) platform[60]. Reads were then mapped to the human reference genome build hs37d5 (phase II reference of the 1000 Genomes Project including decoy sequences) using the Burrows-Wheele Aligner (BWA) version 0.7.15 mem function with default parameters (except for invoking -T 0)[61,62]. Duplicates were marked with Sambamba version 0.6.5[63]. Single-nucleotide variations (SNVs) and insertions/deletions (indels) were called using an in-house workflow, based on SAMtools/BCFtools 0.1.19 (for SNVs) and Platypus 0.8.1 (for indels)[64,65]. The annotation was performed with ANNOVAR (version 2016Feb01)[66]. To remove artifact SNVs/indels we calculated a 'confidence score' for each mutation. This score was first defined as 10, and then deducted if the mutation overlapped with repeats or DUKE excluded regions, DAC blacklisted regions, self-chain regions, or segmental duplication records[67,68]. For indels, the filters from Platypus were additionally considered to calculate the confidence score, which was deducted if alleleBias, badReads, MQ, SC, GOF, QD, or strandBias was set. Thus, mutations were excluded from the analysis if this confidence score was too low (<8), if sequencing depth was too high or too low, or if the reads were not properly mapped. Common polymorphisms that could be found in mutation databases were further filtered out: variants were excluded if present in dbSNP 147 with 'COMMON = 1' tag (although rescued if they had the corresponding OMIM record), the Exome Aggregation Consortium (ExAC) database 0.3.1 (>0.1%)[69], the Exome Variant Server (EVS) ESP6500SI-V2 (>1%), and in our in-house control data set (>2%, among 280 controls). Based on ANNOVAR annotations, only the variants in coding regions were selected for the analysis. The oncoprint plot was finally generated using the ComplexHeatmap R package[70].

**Data availability**. Infinium MethylationEPIC BeadChip data are available from the ArrayExpress database under the accession number E-MTAB-5738. Whole-exome sequencing data has been deposited at the European Genome-phenome Archive (EGA), which is hosted by the EBI and the CRG, under accession number EGAS00001002670.

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

## Acknowledgements

We thank the DKFZ Genomics and Proteomics Core Facility for Infinium 850k and whole-exome sequencing services and the Data Management Group (DMG) for their support in the whole-exome sequencing data analysis. This work was supported by the BMBF-funded German Network for Bioinformatics Infrastructure de.NBI (grant numbers. 031A537A, 031A537C, and 031L0101A). We thank Gregor Warsow for his assistance in preparing the Oncoprint plot and Barbara Hutter for advice.

## Author contributions

M.R.-P., F.B., G.R. and J.G. analyzed the data. C.L.-P. performed the immunohistochemistry experiments. F.K. and E.W. contributed to the data analysis. E.D., L.F., J.P. and B.B. carried out whole-exome sequencing data analysis. K.S. prepared epidermis samples. S.G., H.W., J.R.-H., E.H. and I.N. organized the collection of samples. M.W. and F.L. conceived the study.
M.R.-P. and F.L. wrote the paper. All authors read and approved the final manuscript.

## Additional information

**Competing interests:** E.W., K.S., S.G., H.W. and M.W. are employees of Beiersdorf AG. F.L. has received consultation fees from Beiersdorf AG.

