## [Peer Review File · Nature Communications]

Reviewers' comments:

Reviewer #1 Expert in cSCC:

The article by Dr Lyko and colleagues interrogates human skin, actinic keratoses and cutaneous SCC for genomic methylation profiles. The authors provide data that suggests the methylome of AK is very similar to SCC but different from unremarkable skin. However, it appears that this is accurate in about 50% of cases. The topic of this study is important and could provide novel insights regarding methylome changes during UV-induced carcinogenesis. However, the level of experimental rigor is not clearly explained. Also there are some issues with harvesting of AK lesional cells that needs clarification.

The following issues need to be addressed.

1. Intro-the authors do not clearly state the hypothesis they are testing. This should be addressed as genomic techniques such as methylome analysis are hypothesis generating studies rather than hypothesis testing studies. Alternatively, is this just a descriptive study to describe the methylome of skin samples, AKs and SCCs.
2. There is a significant issue with procurement of the AK tissue. As described, it would not be possible for the authors to conclude that these 4 mm punch biopsies did not contain squamous cell carcinoma in situ with the actinic keratosis. This distinction could only be done if the authors performed laser capture microdissection. Clinically it is difficult to distinguish AK from SCCIS and very frequently they occur together in the same punch biopsy. The author may have to state that the AK samples may have contained SCCIS.
3. The PCA analysis and fact that there are so few differences between AK and SCC raises the possibility of SCCIS in the AK samples. Biologically AK and SCC are very different so if the methylome is so similar why so little difference in the methylome.
4. In fig 4, the authors state that the p63 immunohistochemical staining in healthy skin is confined to the basal cell layer, the stem cell compartment. The data clearly show p63 staining extending above the basal cell layer to the granular layer. This pattern is typical for unremarkable skin. I am not sure why the authors state that the p63 staining is confined to the basal layer.
5. Regarding the data on keratin promoter methylation, the authors should state that AKs and SCC cells can express keratins typically found in differentiated cells rather than saying AKs and SCCs arise from keratinocytes with varying degrees of differentiation cells. The latter conclusion is not accurate as AK and SCC cells can aberrantly express genes typically found in differentiated keratinocytes. AK and SCC cells, at least the proliferative portions are not differentiated in the sense that epidermal keratinocytes differentiate because they do not withdraw from the cell cycle.
6. The authors show similarity in the methylome profiles but AKs and SCCs are very different lesions both clinically and biologically. As the authors noted AKs have a high rate of spontaneous regression, SCCs do not. Does the data presented mean that the methylome pattern of AKs and SCCs does not correlate with the key biological properties of AKs and SCCs?
7. Are there any novel genes that were identified as being expressed in AKs and SCCs but not healthy skin? If so, can the data be presented and confirmed by immunostaining.
8. The significance of the EpSC like phenotype is not clear as the EpSC cells in AKs and SCCs have a much higher proliferative rate than epidermal stem cells. Using a differentiation scale may be more appropriate.

Reviewer #2 Expert in epigenomics:

The authors have carried out an array-based analysis of DNA methylation patterns in a modestly-sized series of human actinic keratoses and squamous cell carcinomas. Their bioinformatic analysis of the resulting Illumina EPIC array methylation data appears to be technically adequate. They conclude (but see below) that SCCa's may be of two biological classes.

However, the study has two main limitations, which this reviewer believes need to be addressed with more data:

1. The number of samples is simply too small to allow a clear separation of the methylation patterns of the (putative) two classes of AK and SCCa by (non-supervised) Principle Component Analysis. Adding more AK and SCCa samples might lead to a believable separation. More samples should be added to their series.

2. By this point, the "state of the art" for tumor profiling is combined genetic and epigenetic analysis. The authors should run at least a subset, if not all, of their SCCa samples by exome sequencing - which may produce data regarding the spectrum of somatic mutations that will better elucidate the putative two classes of SCCa. Such data should be included in their report, not published separately.

Reviewers' comments:

Reviewer #1:

1. Intro-the authors do not clearly state the hypothesis they are testing. This should be addressed as genomic techniques such as methylome analysis are hypothesis generating studies rather than hypothesis testing studies. Alternatively, is this just a descriptive study to describe the methylome of skin samples, AKs and SCCs?

This point has been clarified at the end of the introduction: "Our results provide the most comprehensive epigenomic analysis of cSCC development to date and suggest the existence of two distinct subclasses that reflect different cell-of-origin differentiation stages."

2. There is a significant issue with procurement of the AK tissue. As described, it would not be possible for the authors to conclude that these 4 mm punch biopsies did not contain squamous cell carcinoma in situ with the actinic keratosis. This distinction could only be done if the authors performed laser capture microdissection. Clinically it is difficult to distinguish AK from SCCIS and very frequently they occur together in the same punch biopsy. The author may have to state that the AK samples may have contained SCCIS.

All AK samples used in this study are taken from the center of the lesions and further dissected to allow detailed histopathological analyses of the core of the lesions. AK samples with histopathological features of SCCIS were excluded from further analysis. This point has now been clarified in the main text and in the methods section.

3. The PCA analysis and fact that there are so few differences between AK and SCC raises the possibility of SCCIS in the AK samples. Biologically AK and SCC are very different so if the methylome is so similar why so little difference in the methylome.

See point 2 above, SCCIS was excluded for all samples by histopathological analysis. Additionally, we have now provided a more detailed explanation in the discussion regarding the similarities between AK and cSCC due to their shared cells-of-origin.

4. In fig 4, the authors state that the p63 immunohistochemical staining in healthy skin is confined to the basal cell layer, the stem cell compartment. The data clearly show p63 staining extending above the basal cell layer to the granular layer. This pattern is typical for unremarkable skin. I am not sure why the authors state that the p63 staining is confined to the basal layer.

We agree with the reviewer and have corrected the text accordingly.

5. Regarding the data on keratin promoter methylation, the authors should state that AKs and SCC cells can express keratins typically found in differentiated cells rather than saying AKs and SCCs arise from keratinocytes with varying degrees of differentiation cells. The latter conclusion is not accurate as AK and SCC cells can aberrantly express genes typically found in differentiated keratinocytes. AK and SCC cells, at least the proliferative portions are not differentiated in the sense that epidermal keratinocytes differentiate because they do not withdraw from the cell cycle.

The corresponding section of the manuscript ("A specific methylation signature at keratin gene clusters identifies two distinct subclasses of AK/cSCC.") describes different methylation patterns of the keratin gene clusters. This provided a first indication for the subtype-specific methylation patterns described further down in the manuscript. We have rewritten the sentences at the beginning of this section to clarify this point. Also, we do not suggest any similarities to non-proliferating (terminally differentiated) keratinocytes. Our further revised model (Fig. 7) illustrates this point.

6. The authors show similarity in the methylome profiles but AKs and SCCs are very different lesions both clinically and biologically. As the authors noted AKs have a high rate of spontaneous regression, SCCs do not. Does the data presented mean that the methylome pattern of AKs and SCCs does not correlate with the key biological properties of AKs and SCCs?

While AKs and SCCs indeed show major clinical differences, they both originate from the same cell type(s). While we acknowledge the differences in our introduction, our results focus on the epigenetic continuity from the cell of origin to the clinical lesion. The correlation between subtype-specific methylation patterns and clinical features will be an important point of future studies (also see point 8 below).

7. Are there any novel genes that were identified as being expressed in AKs and SCCs but not healthy skin? If so, can the data be presented and confirmed by immunostaining.

The identification of novel differentially expressed genes and their confirmation by immunostaining will be an important aspect for future research. This is now mentioned in the text. Importantly, we have now also included whole-exome sequencing data (Fig. S4) to provide more comprehensive molecular profiles of the two AK/cSCC subclasses.

8. The significance of the EpSC like phenotype is not clear as the EpSC cells in AKs and SCCs have a much higher proliferative rate than epidermal stem cells. Using a differentiation scale may be more appropriate.

The significance of the two subtypes will be a key aspect for future studies. This is now mentioned in the text. Also, we consistently distinguish between EpSCs and "EpSC-like"

cells to emphasize potential differences. This is also reflected in the revised version of our model (Fig. 7).

Reviewer #2:

1. The number of samples is simply too small to allow a clear separation of the methylation patterns of the (putative) two classes of AK and SCCa by (non-supervised) Principle Component Analysis. Adding more AK and SCCa samples might lead to a believable separation. More samples should be added to their series.

Our analysis is now based on 16 AK and 18 cSCC samples, thus almost doubling the numbers of the original submission. The additional datasets are entirely consistent with our original findings and thus provide important confirmation for our original conclusions.

2. By this point, the "state of the art" for tumor profiling is combined genetic and epigenetic analysis. The authors should run at least a subset, if not all, of their SCCa samples by exome sequencing - which may produce data regarding the spectrum of somatic mutations that will better elucidate the putative two classes of SCCa. Such data should be included in their report, not published separately.

Whole-exome sequencing was performed for 10 AK and 10 cSCC samples (Tab. S2). The mutational profiles are consistent with the published literature and are shown in Fig. S4. Correlations between mutational profiles and methylation subtypes could not be detected (Fig. S4), which provided important further support for our main conclusion, i.e. that DNA methylation subtypes are defined by the cell of origin.

REVIEWERS' COMMENTS:

Reviewer #1 (Remarks to the Author):

The authors have addressed the issues raised in a reasonable manner.